# The Effect of Vinyasa Yoga Practice on the Well-Being of Breast-Cancer Patients during COVID-19 Pandemic

**DOI:** 10.3390/ijerph20043770

**Published:** 2023-02-20

**Authors:** Agnieszka Zok, Monika Matecka, Joanna Zapala, Dariusz Izycki, Ewa Baum

**Affiliations:** 1Department of Social Sciences and Humanities, Poznan University of Medical Sciences, 60-812 Poznan, Poland; 2Division of Philosophy of Medicine and Bioethics, Poznan University of Medical Sciences, 60-812 Poznan, Poland; 3Department of Occupational Therapy, Poznan University of Medical Sciences, 60-812 Poznan, Poland; 4Department of Postgraduate Studies, SWPS University, 03-815 Warszawa, Poland; 5Department of Cancer Immunology, Poznan University of Medical Sciences, 60-812 Poznan, Poland

**Keywords:** life quality, yoga, insomnia, well-being, physical activity

## Abstract

Background: Vinyasa yoga practice improves body fitness and potentially positively affects practitioners’ well-being and health. Due to the diverse intensity of practice and positions customized to the practitioner’s needs, it can also support cancer patients. Undertaking physical activity that has a potentially positive effect on well-being and health was particularly important during the self-isolation that followed the COVID-19 pandemic. The purpose of this study was to evaluate the impact of three-month mild and moderate intensity vinyasa yoga practice on breast-cancer patients’ stress perception, self-confidence, and sleep quality during COVID-19 induced self-isolation. Methods: Female breast-cancer patients participated in twelve-weeks of online vinyasa practice during the COVID-19 induced self-isolation period. Meetings were held once a week, where 60-min vinyasa yoga sequences were followed by 15 min of relaxation. Patients completed pre- and post-intervention surveys to evaluate changes in the following outcomes: stress perception, self-confidence, and sleep quality. Forty-one female patients enrolled in the Vinyasa course completed the pre-intervention survey, while 13 attended all the meetings and completed the post-intervention survey. Results: The effect of the twelve-week yoga and relaxation practice significantly reduced sleep problems and stress of oncological patients. The participants also declared an improvement in their general well-being and self-acceptance. Conclusion: Dynamic forms of yoga combined with mindfulness techniques can be applied to patients treated for oncological diseases. It contributes to improving their well-being. However, in-depth studies are needed to analyze the complexity of this effect.

## 1. Introduction

Physical activity evokes growing interest in cancer patient rehabilitation [1]. An increasing number of studies have shown that low to moderate-intensity exercises could help cancer patients to recover, deal with treatment, and also have an impact on extending survival time [2,3,4,5]. It is worth mentioning that physical activity leads to improved physiological and psychological functioning in oncological patients (a rehabilitation of patients with advanced cancer). The importance of physical activity in the prevention [6] and supportive treatment [7,8] of oncological diseases is increasingly emphasized. The positive impact of physical activity in the period of recovery from the illness has also been observed [6]. Despite this, patients avoid activity, and engaging patients in physical activity is challenging for healthcare professionals [9].

Patients are also recommended various forms of psychotherapy [10,11,12], such as relaxation and guided imagery [13,14], mindfulness meditation (mindfulness-based stress reduction) [14], music therapy [15,16], occupational therapy [17,18,19], and breathing exercises [20,21,22]. Most of the mentioned therapies are a part of vinyasa yoga training, which always includes breath work (*Pranayama*) and relaxation (*Shavasana*), enhanced by aromatherapy, music therapy, or/and visualization techniques or Nidra yoga. The sequence of vinyasa yoga is created individually by the instructor (in contrast to, e.g., ashtanga yoga), which enables adjusting the exercises to the patients’ individual needs. Therefore, it is assumed that the practice of vinyasa yoga can be a form of support for patients with chronic diseases, including oncology, in treatment and return to physical activity, thus affecting the improvement of psychological well-being. It should also be remembered that the project was carried out during the COVID-19 pandemic when many patients with chronic diseases were deprived of access to rehabilitation and social contacts were significantly reduced. This is important because it is rooted in a social network and the resulting ability to receive informal support that significantly impacts a person’s well-being [23]. Physical activity was also strongly associated with a reduced risk of severe COVID-19 complications among infected adults [24,25,26].

During the COVID-19 pandemic, in the period covered in presented study, self-isolation was the recommended way to prevent SARS-CoV-2 infection. Self-isolation was extremely important, especially for cancer patients, as they are a group at higher risk of death or complications due to their systemic immunosuppressive state caused by the disease and its drug treatment [27]. In addition, cancer patients tend to be older and have other comorbidities (e.g., hypertension, diabetes, and cardiovascular disease) associated with a worse prognosis in COVID-19, reinforcing the importance of self-isolation efforts in these patients [28]. A negative consequence of self-isolation was physical inactivity and prolonged sitting, as well as increased anxiety, depressive symptoms, and increased fatigue [28].

The study aimed to analyze the effect of yoga intervention complemented by relaxation exercises on the well-being of patients treated for oncological diseases during the COVID-19 pandemic. Special attention was paid to the aspects of insomnia and lowered mood, which may have been exacerbated by the need for isolation. Patients’ expectations of starting exercise, and the recognized effects of exercise, were also examined.

## 2. Materials and Methods

The study was approved by the Bioethics Committee at the PUMS No. 181/21.

### 2.1. Patients Recruitment

Breast-cancer patients were recruited through the “Otulina” and “PsycheSomaPolis” NGO foundations providing wide and comprehensive care for oncology patients (https://fundacjaotulina.pl (accessed on 4 January 2022), https://www.psychesomapolis.org (accessed on 4 January 2022)). A total of 41 breast-cancer patients were recruited from January to April 2021. Inclusion in the study occurred through an interactive online registration form. During registration, patients were asked to complete pre-intervention surveys (The survey form and row data answers are attached as Appendix A). After completing the series of yoga sections, the patients were asked to respond to post-intervention surveys. Participation was entirely voluntary. Participants were instructed that they could withdraw from the study at any time without giving a reason in accordance with the applicable ethical rules.

### 2.2. Questionary Form

The survey was prepared using Google Forms. Each patient received a link to the survey and filled it out on their computer. The survey questionnaire contained three sections. In the first, respondents answered questions related to their current problems with the disease and their experiences with yoga practice. The second section of the questionnaire dealt with issues related to the patient’s quality of life with particular attention to self-perception and level of independence. Questions in this section addressed issues related to the daily activities of cancer patients and their levels of happiness/sadness. The last part consisted of sociodemographic questions, where we asked about age, place of residence, or education. Respondents completed the questionnaire before and after the project. The final questionnaire included additional questions about the results of the classes. The post-intervention questionnaire was performed directly after the last meeting with the BC patients that had participated in all meetings.

### 2.3. Intervention and Yoga Exercise Protocol

Vinyasa yoga sessions were conducted from April to the end of June 2021 every Wednesday at 4 p.m. Due to the COVID-19 self-isolation periods, the practices were carried out online via the Zoom application. Each participant had their own computer with camera, internet access, their own exercise mat and a home-prepared space for performing exercises. During the meetings, the instructor explained the various positions and performed them with the participants. The 60-min vinyasa practice was followed by fifteen-minute relaxation (Shavasana).

The vinyasa postures (asanas) were carefully selected considering the individual needs and abilities of the patients and adapted to their health condition. At the beginning, a personal health report was collected regarding the patients’ condition, complications and possible contraindications to the exercises. Appropriate exercise sequences were created to increase the accuracy of the selection of exercises and prevent possible complications. Yoga activities were also adapted to the patients after lymphadenectomy. Therefore, transitions in yoga postures were limited to 4-point kneeling with appropriate cues and “cat” and “cow” poses (viniasa positions). The practice began in a comfortable sitting position (on a mat or chair), calming the breath and performing simple exercises to mobilize the available body part. The second part featured more vigorous exercises involving different parts of the body. Positions that required lifting the arms and opening the chest (such as *gomukmhasana*) were performed. Attention was focused on equivalent positions, such as tree pose (*vrikasana*) or warrior 3 in the flowing sequence of vinyasa Yoga. The third part was dominated by lying down positions, using aids such as bolsters, blocks, and straps (the aids could be used at any time during the class). The last position was *shavasana*. Relaxation was performed lying on the back with arms spread wide and eyes closed. During relaxation, the instructor involved the following techniques: bibliotherapy, music therapy, mindfulness rotation training, and visualization.

### 2.4. Statistic

Pre- and post-intervention survey results were downloaded and saved in “xlsx” format. Statistical analysis and visualization of the obtained results were performed in the statistical software environment “R” (version 3.5.1) with several additional libraries. Xlsx files were imported into R using the “openxslx” library [29]. The number of response options given to the respective questions and the corresponding percentages were calculated. The obtained values were shown as a table or were visualized using the ggplot2 library [30].

In the next step, we analysed the responses submitted before and after the intervention. Using the personal ID included in both surveys, data were extracted for only patients who fulfilled pre- and post-intervention surveys. The Shapiro-Wilk normality test was used to verify the data’s conformance to a normal distribution. Since most of the data did not meet the assumptions of a normal distribution, a non-parametric test was used for statistical analyses. The Wilcoxon signed-ranks test was used for group comparisons. The values of *p* < 0.05 were considered statistically significant. The median, interquartile range, mean, and standard deviation were calculated and presented on the graphs.

### 2.5. Analysis of Multiple Answer Questions

The total number of responses, the percentage per response, and the percentage per patient population were calculated and presented as a table using the “multiResponse” function from the “userfriendlyscience” package [31]. Correlations between the given answer variants were visualized using the “arcdiagram” package [32]. The obtained arc plot showed the level of correlation representing the exponent of the answer variants most frequently marked together. Individual answers in multiple-choice questions were also presented by means of heatmap where the answer to a given variant was marked in black colour. For this purpose, the “Complexheatmap” package was used [33].

## 3. Results

Forty-one female cancer patients (N = 41) joined the project and solved the initial survey form. Thirteen patients completed the full cycle of yoga practice and fulfilled the final survey (N = 13). The socio-demographic characteristics of the group regarding the answers given before starting the practice are shown in Table 1.

More than 65% of female participants (N = 27, 65.85%) admitted that they practiced yoga occasionally. Only one person declared regular practice (N = 1, 2.44%). A total of 31% had never practice yoga in any way before (N = 13, 31.71%).

More than half of the participants (N = 22, 53.6%) completed treatment with chemotherapy and radiotherapy. During the project, 16 patients (39%) underwent treatment, and three declared themselves ineligible for systemic therapy.

The largest group of patients were women between 37–45 (N = 13, 31.7%) and 46–51 (N = 12, 29.3%) years of age. Most of the women said they live with their families (N = 24, 58.4%). Residents of large cities (N = 20, 48.8%) with higher education (N = 34, 83%) predominated among the subjects.

In the first part of the study, we analyzed the health problems and ailments of breast cancer patients who participated in yoga classes. Figure 1 shows the results for the pre-intervention survey. A comparison of the answers given before the beginning of the course with those answered after the final meeting of vinyasa yoga course was also carried out and is presented as Appendix A. The analyzed question set included only answers from a numerical scale, where 1 means never, and 4 means very often. In order to identify the most common health problems and ailments, the questions set were ranked from those with the highest number of responses answering 4. Consistent with this approach, patients very often felt irritable, worried, and depressed in the past week (respectively 26.83%, 26.83%, 21.95% very often and 34.15%, 21.95%, 19.51% often). The patients also declared sleep difficulties in the past week (21.95% very often, 21.95 often). On the other hand, a short walk outside the home never caused problems for most patients (68.29%).

We also examined patients’ motivations to practice vinyasa yoga. One of the survey questions completed before the course—“What are your motivations to practice yoga?” was a multiple-choice question containing the most common reasons for starting the practice. The results are shown in Figure 2. Most often, patients indicated that the reason for joining the course was the desire to stretch the body (N = 32), the need to increase vitality (N = 29), and better stress management (N = 26). According to the correlation analysis of the provided answers, presented on the arc diagram (Figure 2B), the patients mostly marked together “the need to increase vitality”, “the desire to stretch the body”, and “better stress management”. Percentage of responses (in relation to total frequency = 157) and percentage of (41) cases (percentage of response frequency in relation to 41 patients) are presented on Figure 2C.

In the next step, we analyzed changes in patients’ responses before and after the intervention. For this purpose, we selected sets of responses assigned to individual patients from two questionnaires. Figure 3 shows the changes in responses regarding irritability, depression, difficulty sleeping, and loss of self-confidence. The questions were answered on a point scale where 1 means never while 4—very often. The questions were formatted in such a way that a lower score indicates a reduction in the analyzed problem. Thus, the potential benefit of the intervention for patients was expressed by lowering the mean score values. It was observed that female patients declared a statistically significantly lower score value in the “Have you experienced difficulty sleeping in the past week?” question after completing the vinyasa yoga course (mean before intervention = 2.46, mean post intervention = 1.77), indicating that their problems related to difficulty sleeping decreased. The patients’ feelings of depression were statistically lower after the intervention (mean pre-intervention = 2.46, mean post-intervention = 2.08). Patients’ sense of irritability was significantly lower after the intervention (mean pre-intervention = 2.69, mean post-intervention = 2.15). We also showed that self-confidence increased significantly in breast cancer patients after the vinyasa yoga course. The mean of the responses to the question “Have you recently lost self-confidence?” was significantly lower in the responses given after the course (mean pre-intervention = 2.08, mean post-intervention = 1.69).

After three months of exercise, the participants’ satisfaction levels improved significantly.

To analyze overall efficiency and satisfaction, we compared participants’ responses to two questions. Figure 4 shows the changes in responses regarding ability to perform daily activities and subjective assessment of one’s own health. The survey questions were answered regarding a numerical scale where, for Figure 4A: 1 means more than usual and 4—less than usual. It was observed that the patients declared a statistically significantly lower score on the question “Have you been able to enjoy your usual daily activities lately?” after completing the vinyasa yoga course (mean before the intervention = 2.15, mean after the intervention = 1.69), indicating that their problems with performing daily activities had decreased.

Figure 4B: presents changes in patients’ perceptions of their own health, where 1 means very bad and 7—perfect. We observed that female participants declared a statistically significant higher score on the question “How do you assess your overall health during the past week?” after completing the project (mean before the intervention = 4.85, mean after the intervention = 5.62), indicating that they assess their health status as better.

Another set of questions was analyzed to assess the satisfaction and depressive tendencies of female patients. Cancer patients chose one of the individual statements that most characterized their physical and emotional state, determined before and after a 3-month yoga practice. The analysis showed that exercise positively affected the patients’ mental state. Before the start of the project, 46.15% of participants answered affirmatively to the question “I’m not sad or depressed” after completion, 61.54% gave such an answer. Participants look to the future more positively. To the question “I often worry about the future” before the start of the class, 53.85% answered affirmatively, after the affirmative answers were 38.46%. According to the participants’ declarations before the project, 30.77% answered the question “I’m not too worried about the future” affirmatively, while after, 53.85% declared no worries. Self-satisfaction, on the other hand, was declared by 53.85% of participants starting the project and 61.54% upon completion. However, the patient’s tendency to cry decreased. The question “I don’t cry more than usual” was answered in the affirmative by 53.85% of people. After the project, the responses were 92.31%. Participants declared a lesser tendency toward nervousness after taking the course. The question “I’m more nervous and unpleasant than before” was answered affirmatively by 38.46% of participants before the exercise, and after by only 7.69%. In addition, participants indicated that the project significantly improved the quality of their sleep, with 61.54% saying that they slept better after the project than before; before the project, good sleep was reported by 15.38%. A total of 84.62% of those starting the project answered in the affirmative to the question, “I’m tiring much faster than before”, while after the exercise, 46.15% declared tiring quickly.

At the end of the project, we asked the participants to declare the effects of the exercises they noted. In response to the question, we gave the trainees multiple choices. The most frequently chosen answer was “Improve body flexibility,” declared by ten patients; nine chose “stress relief.” The answer “Improving mental health” was selected by eight participants.

## 4. Discussion

The coronavirus disease (COVID-19) pandemic has radically changed our daily activities. To reduce the pace of the pandemic, the World Health Organization has recommended social isolation measures, especially in groups at risk of severe infection, such as the elderly and people with chronic illnesses, including cancer patients [34]. Since isolation precautions also apply to sports activities, home workouts remained the only option for playing sports and staying active during the pandemic due to the introduced restrictions, and the patients’ physical activity decreased significantly [35]. However, a growing number of studies indicate a significant influence of appropriately selected physical activity on the healing process [9,36]. The effectiveness of the online exercise protocol was confirmed by a study by Grazioli et al. indicating that training conducted remotely can also help improve patients’ quality of life [37]. In comparison, the effectiveness of combined training in a non-cancer population is indicated by a meta-analysis by Jamka et al., who showed a more favorable effect of such an intervention on glucose and insulin homeostasis and lipid profile [38]. The effectiveness of home-based intervention during a pandemic was also confirmed by Natalucci et al., who indicated that a 3-month home-based lifestyle intervention focused on a Mediterranean diet and aerobic exercise significantly reduced echocardiographic signs of diastolic dysfunction and improved autonomic function. In addition, there were significant improvements in BMI, cardiorespiratory fitness, metabolic and inflammatory parameters [39]. Our study also confirmed that online classes could both improve patients’ fitness and positively affect their psychological well-being (Figure 5). Scientific evidence supports the safety and effectiveness of physical activity, indicating—for people with cancer experience—health benefits such as reduced anxiety, reduced depressive symptoms, reduced fatigue, improved quality of life, and improved physical function [40,41,42,43]. The importance of sleep and relaxation for both mental and physical health is also emphasized. This is particularly relevant during the pandemic period, as isolation also induced feelings of loneliness among patients, which can further compound negative health effects not only in terms of mental health, but also with physical illnesses such as cardiovascular disease [44,45] and elevated blood pressure [46,47]. In addition, a systematic review noted that loneliness and social isolation were risk factors for early mortality [48,49,50]. Hawkley et al. suggest that loneliness among middle-aged and older adults is an independent risk factor for physical inactivity and increases the likelihood of physical inactivity cessation over time [51]. Researchers point to the key role of social support in initiating physical activity and regularity [52,53]. Their findings provide an important argument for encouraging exercise among people at risk of loneliness due to isolation. Although online classes do not solve the problem of loneliness, they provide motivation to exercise and reinforce the regularity of the effort taken.

The analysis by Zapala et al. identified several needs of oncology patients during the pandemic. The authors indicated that the desire for recovery is not the only need of patients. Peace of mind significantly influences their well-being, which can be achieved through physical activity and breathing exercises. The pandemic has significantly complicated and hindered the process of identifying and adequately responding to the needs of cancer patients. Consequently, it has exacerbated the suffering of patients and their relatives [54]. Therefore, the activities we conducted responded to the needs of patients and resulted in improved satisfaction with daily life activities and assessment of overall health (Figure 4).

Although the positive effects of yoga on patients’ psychological well-being have long been known [55,56,57,58,59] the study conducted in this project showed that cancer patients could successfully practice vinyasa yoga. Modern yoga’s distinctive blend of movement and breath maximizes the benefits of physical activity. During practice, patients not only strengthen and stretch their body but also improve their breathing capacity [60]. Related to the practice, learning calm, prolonged breathing temporarily improves patients’ well-being [61], and teaches them to cope with the stress they may experience in the future. Measures to reduce stress appear to be extremely important, as many studies have shown that stress promotes the development of cancer [62,63,64,65,66]. Furthermore Chalaye et al., observes that Slow Deep Breathing is a simple and easy-to-use method of relieving acute pain that can easily be used during painful medical procedures to alleviate an acute pain crisis or as a complementary pain therapy for chronic pain [67]. Busch remarks that prolonged calm breathing induces a decrease in sympathetic activity and attenuation of pain perception, suggesting that breath intervention is easy to learn [68]. Researchers note that elements of physical, mental, and spiritual disciplines such as most styles of yoga, Qi-Gong, and Tai Chi can be used to treat chronic bulbar conditions [68,69,70,71]. However, previous researchers rarely indicated the style of yoga training. The most common practice was the method of Iyengar [72,73,74,75], according to which the postures are worked statically with the use of many aids (chairs, rollers, blocks, straps); after taking one position, the practitioner maintains it and refines it with micro-movements. The practice is devoid of dynamic transitions between positions. It is definitely a mild form of physical activity. Vinyasa yoga is characterized by much greater equanimity. It can take the form of mild or moderate activity. In addition to the effects of yoga associated with mindfulness practice, systematic participation in Vinyasa yoga can effectively improve cardio-respiratory fitness and strengthen the body and can be used as an alternative method to traditional aerobic exercise [76]. *Vinyasa* is a dynamic style that emphasizes the seamless connection between movement and breath. The dynamic nature of vinyasa yoga helps strengthen the body further and improve the student’s condition [77,78]. The freedom to create sequences and the lack of need for aids allow for a customized practice that can be done almost anywhere, making it easier for patients to engage in this activity. Patients participating in our project were able to perform elements of the practice even during hospitalization. A piece of floor, a carpet/mat, or socks to prevent slipping were sufficient. In addition, the combination of mild to moderate vinyasa yoga training and breathing practice benefits health holistically, increasing physical fitness and facilitating a sense of relaxation and inner peace [79]. The awareness of breathing, which is the basis of yoga, and the dynamics characteristic of vinyasa influence the psychological well-being of the patients and improve their physical efficiency, which results in a reduction in fatigue during longer distances (walking). Pina et al. note that even a single session of vinyasa yoga can have a positive effect on the mood of exercisers. Researchers highlight the efficacy of a single bout of vinyasa yoga in improving AIx, an index of wave reflection influenced by vascular resistance and systemic arterial stiffness [77,80]. In addition, note the effectiveness of vinyasa yoga in enhancing mood and non-HDL cholesterol levels [77]. The results of our research confirm the positive effects of vinyasa on well-being. It has been confirmed both in the aspect of reduction in fatigue and better rating of current well-being (Figure 3 and Figure 4).

Insomnia is a common problem among cancer patients, potentially devastatingly impacting their quality of life [81,82,83,84]. Despite its prevalence, oncology literature rarely considers it [81,85,86,87,88], and patients can rarely count on ongoing systemic assistance.

Patients at the beginning of the project complained of severe sleep problems (Figure 2). Multiple factors contribute to insomnia among patients with breast cancer, including endocrine therapy and hot flashes, pain and discomfort from local therapy, and fear of recurrence [81,87]. The SRAS-CoV-2 pandemic further compounded the problem of insomnia during our research. The pandemic has contributed to health problems such as stress, anxiety, depressive symptoms, insomnia, denial, anger, and fear on a global scale [89,90,91]. The age of our female patients can also be considered a factor that increases the risk of insomnia [92]. Therefore, improving sleep quality in female patients appears to be a meaningful result. Those participating in the supervised project reported a significant reduction in sleep problems after a series of yoga practices (Figure 3).

The effectiveness of non-pharmacological support for insomnia is also suggested by Zeichner et al., noting that such intervention can reduce comorbidities and reduce healthcare resource utilization. Cognitive-behavioral therapy for insomnia, mindfulness, and yoga are three behavioral health interventions they recommend for treating sleep disorders in cancer patients [93]. A meta-analysis conducted by researchers in Taiwan on the effects of yoga on sleep quality indicated—surprisingly enough—an improvement in sleep quality in female patients who followed the practice [94]. However, it should be noted that previous studies did not take into account such important aspects as the style of yoga practiced by the participants of the classes and the time of day when the classes were held. Meanwhile, when analyzing the relationship between yoga practice and sleep quality, it is essential to consider a mediating factor—the type of asanas (postures) performed at a particular time of day. Intense evening practice, combined with *ujjayi* breathing [95] is very energizing, which may adversely affect sleep parameters. Taking this into account, the patients participating in the self-project were offered classes in the morning. Calming breathing such as box breathing, *Nadi shodan* [96], or *bhramari* [97,98] were also used at the end of the class. Since the yoga sessions were recorded and additional breathing exercises were prepared for the patients, they could apply them at any time.

As declared by project participants, a reduction in stress, feelings of depression, and nervousness testify to an improvement in the subjects’ quality of life. Researchers from Shiraz University, who conducted an eight-week yoga course for women with breast cancer-related lymphedema, came to similar conclusions. They noted that Yoga exercises could influence most of the functional and symptom-related aspects of QoL in women with breast cancer. A significant increase in physical, cognitive, and emotional functioning and a significant reduction in pain, fatigue, insomnia, and financial difficulties should be noted as the positive result of yoga exercise [99]. Zhi et al. note that yoga may reduce feeling in patients with chemotherapy-induced peripheral neuropathy [100]. Although the researchers point to the need for additional studies, our patients’ feelings of anxiety decreased significantly.

The improvement in subjective feelings of tension, stress, and health anxiety that we have observed in patients, may also be related to the fatigue reduction observed by J.E. Bower et al. The analyses showed that yoga intervention targeted at improving fatigue might be a feasible and effective treatment for breast cancer survivors with persistent cancer-related fatigue [57]. A meta-analysis by Essen and London researchers led to similar conclusions, suggesting the need to bolster the evidence [61,101]. P. S. Sahni et al. showed that yoga practitioners perceived greater personal control, greater consistency/understanding, less emotional impact, less risk, and greater preventive control over COVID-19 infection than other individuals. They also showed that physical and mental health patients of yoga practitioners felt that their practice was an effective therapy in coping with COVID-19 [102]. Similar observation was also confirmed in our study (Table 2, Figure 3, Figure 4 and Figure 5).

Although researchers have considered various styles of yoga, vinyasa, due to its fluid, dance-like quality, can deepen feelings of satisfaction with the practice. Vinyasa yoga is called meditation in motion because it focuses on the fluidity of movement, body, and breath. Calming the breath in a comfortable seated position, activating the deep muscles, and protecting the spine—the three elements with which the practice begins—facilitate the focus during class. The fluidity and dynamics of the sequence allows participants to focus on the exercises. The relaxation at the end of the course provides a rest after the exercises, making it easier to return to daily activities. The protocol according to which the classes were conducted took into account all aspects of vinyasa yoga training. Embracing yoga as a whole-physical activity combined with mindfulness techniques can have a tangible impact on the perceived quality of life. This does not involve the spiritual aspect of the practice, which may or may not necessarily accompany the practice of vinyasa, but the purely physical benefits of movement, conscious breathing, and guided relaxation, as we have shown in our research.

### Limitations of the Study

Despite the fact that both our results and the observations of other researchers demonstrate a positive aspect of yoga practice that does not carry the risk of side effects, the study’s limitations must be kept in mind. Our study had several limitations. First, we enrolled a relatively small number of participants. Only women participated in the recruitment conducted by NGOs serving cancer patients. It will be important to expand this study to a larger patient population and determine whether the effects can be generalized to a broader group of people with other cancers. In our study, we did not investigate the long-lasting effect of yoga practice, which seems to have had a huge impact on BC patients’ well-being. In the current study, the core of our methodological approach was based on the pared analysis of BC patients survey answers, before and after yoga practice, however including additional groups, mainly the group of BC patients that did not follow the yoga protocol, and comparing the responses between practising and non-practising BC patients would certainly provide many valuable results. Because the practice of yoga, through its focus on movement and breathing, as well as relaxation, can be categorized as a mindfulness technique, creating a reliable control group, for each aspect of yoga, is difficult.

## 5. Conclusions

Participants in the project, listing their expectations for exercise, indicated primarily a desire to improve vitality, stretch the body, and calm stress. All expectations were met, as well as reducing the problems with sleeping those patients indicated in the questionnaire.

An appropriately sequenced vinyasa yoga can be used successful with patients experiencing cancer illness. Vinyasa yoga helps to manage the problematic psychological effects of therapy, such as anxiety, stress, and tension. Vinyasa yoga is also an effective intervention aimed at improving patients’ fitness. Both breathing exercises and the appropriate intensity of practice and muscle awareness (bandh) contribute to patients being more active in other areas outside of their practice. Yoga practice also alleviates the discomfort of isolation due to the COVID pandemic-19. With the increasing number of infections associated with COVID wave IV, it is worth recommending online vinyasa yoga classes to patients to improve their psychosocial and spiritual well-being and maintain physical activity. It should also be noted that the mechanism of action of yoga practice on the human body is not yet fully understood and requires further study.

## Figures and Tables

**Figure 1 ijerph-20-03770-f001:**
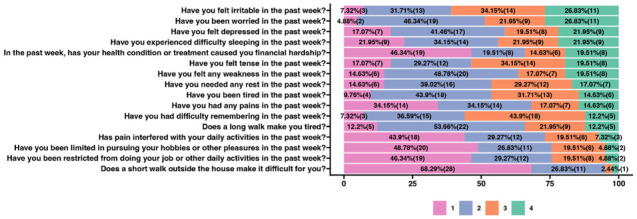
Health problems and ailments of the cancer patients who participated in yoga classes. The number of declared responses (n—shown in brackets) and the percentage distribution (%) were determined based on the results of a survey conducted before the yoga course began. The survey questions were answered with reference to a numerical scale where 1 means never, 2—sometimes, 3—often, and 4—very often. The questions were ordered according to the answers in which most respondents marked 4—as very often.

**Figure 2 ijerph-20-03770-f002:**
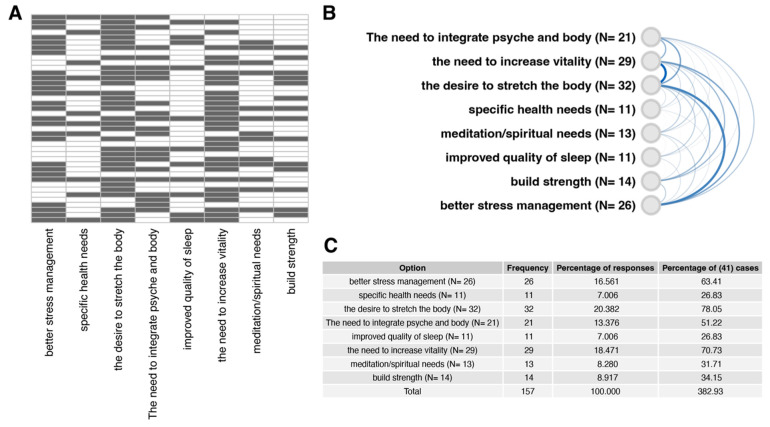
Cancer patients’ motivations to practice yoga. Patients were given the opportunity to indicate multiple or no reasons for participating in yoga practice (multiple answer question). (**A**) a heatmap representing individual responses to the question “What are your motivations to practice yoga?”. The answers of each patient are presented in separate rows (N = 41). The black color of the heatmap cells indicates that the patient’s selection of a particular answer variant. (**B**) Correlation of answer variants aimed at identifying the most frequently selected answers together. The level of correlation was indicated by the intensity and thickness of the line connecting the given answer variants. (**C**) Summary table showing the percentage of responses (in relation to total frequency = 157) and percentage of (41) cases (percentage of response frequency in relation to 41 patients).

**Figure 3 ijerph-20-03770-f003:**
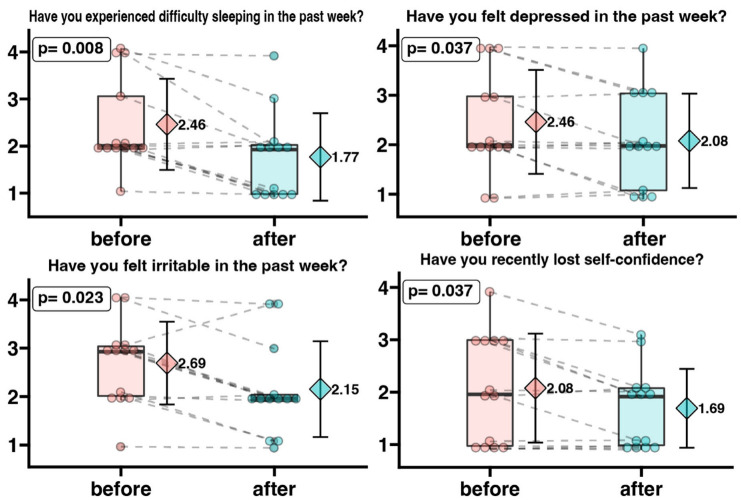
Irritability, depression, difficulty sleeping, and loss of self-confidence level in oncological patients before and after a 3-month yoga practice. The survey questions were answered regarding a numerical scale where 1 means never and 4—very often. Each dot represents an individual patient response. Grey dotted connecting line indicates same patient before and after yoga intervention. The box-plot shows the median and interquartile range. On the right side of each graph, the diamond symbol shows the mean value with standard deviation. A decrease in the mean value indicates a reduction in the analyzed problem.

**Figure 4 ijerph-20-03770-f004:**
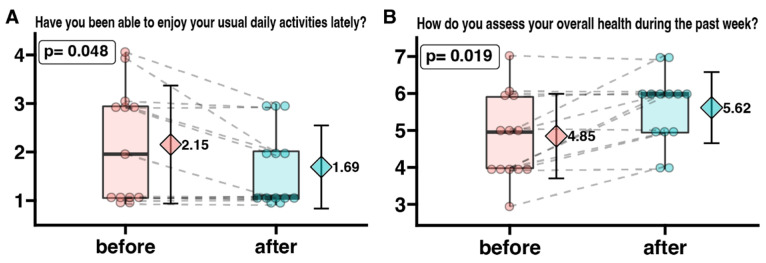
The degree of enjoyment of daily activities (**A**) and patients’ self-assessment of their overall health status (**B**) in oncological patients before and after a 3-month yoga practice. The survey questions were answered regarding a numerical scale where, for Figure 4A: 1 means more than usual and 4—less than usual, and for Figure 4B: 1 means very bad and 7—perfect. Each dot represents an individual patient response. Grey dotted connecting line indicates same patient before and after yoga intervention. The box-plot shows the median and interquartile range. On the right side of each graph, the diamond symbol shows the mean value with standard deviation.

**Figure 5 ijerph-20-03770-f005:**
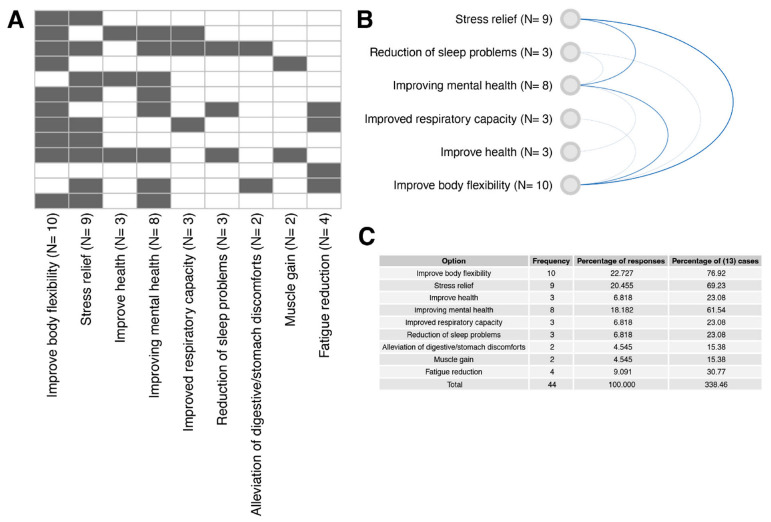
Self-evaluation of yoga practice effects. Patients were given the opportunity to indicate multiple or no benefits after finishing the whole yoga course (multiple answer question). (**A**) a heatmap representing individual responses to the question “Did you notice the effects of yoga practice?”. The answers of each patient are presented in separate rows (N = 13). The black color of the heatmap cells indicates that the patient’s selection of a particular answer variant. (**B**) Correlation of answer variants aimed at identifying the most frequently selected answers together. The level of correlation was indicated by the intensity and thickness of the line connecting the given answer variants. (**C**) Summary table showing the percentage of responses (in relation to total frequency = 44) and percentage of (13) cases (percentage of response frequency in relation to 13 patients).

**Table 1 ijerph-20-03770-t001:** Socio-demographic characteristics of the cancer patients who participated in yoga classes. The number of declared responses (N) as well as the percentage distribution (%) were determined based on the results of a survey conducted before the yoga course.

Parameter	N	%
Age ranges		
25–30	2	4.88
31–36	1	2.44
37–45	13	31.71
46–51	12	29.27
52–60	6	14.63
61–69	7	17.07
Sex		
Women	41	100
I live with		
with the family	24	58.54
with soul mate	11	26.83
alone	6	14.63
Residence		
city with a population of over 100,000	20	48.78
city of 26,000–100,000 residents	5	12.2
city of 10,000–25,000 residents	7	17.07
city of less than 10,000 inhabitants	4	9.76
village	5	12.2
Education		
higher education	34	82.93
high school education	7	17.07
Do you practice yoga?		
Yes, regularly	1	2.44
Yes, sometimes	27	65.85
I never practiced yoga	13	31.71
Are you receiving treatment with chemo/radiotherapy?		
Finished treatment	22	53.66
I am in the process of treatment	16	39.02
I do not qualify for treatment	3	7.32

**Table 2 ijerph-20-03770-t002:** Question sets in which oncology patients chose one of individual statements that most characterize their physical and emotional state determined before and after a 3-month yoga practice. The table presents individual statements with the total number of answers (N) and the percentage distribution. The analysis was performed for patients who completed the entire yoga practice program (N = 13).

Answer	Before (N)	After (N)	Before %	After %
I am constantly so sad and unhappy that it is unbearable	1	0	7.69	0
I am experiencing constant sadness, depression and I cannot free myself from these experiences	1	1	7.69	7.69
I often feel sad and depressed	5	4	38.46	30.77
I’m not sad or depressed	6	8	46.15	61.54
I often worry about the future	7	5	53.85	38.46
I’m afraid that nothing good awaits me in the future	2	1	15.38	7.69
I’m not too worried about the future	4	7	30.77	53.85
I feel self-loathing	1	0	7.69	0
I’m not satisfied with myself	5	5	38.46	38.46
I’m satisfied with myself	7	8	53.85	61.54
I cry more often than I used to	6	1	46.15	7.69
I don’t cry more than usual	7	12	53.85	92.31
Everything that used to irritate me has become indifferent	1	0	7.69	0
I am constantly nervous and irritable	1	1	7.69	7.69
I’m more nervous and unpleasant than before	5	1	38.46	7.69
I’m no more nervous than I used to be	6	11	46.15	84.62
I sleep well, as usual	2	8	15.38	61.54
I sleep worse than I used to	7	5	53.85	38.46
I wake up a few hours too early and can’t get to sleep	1	0	7.69	0
In the morning, I wake up 1–2 h too early and find it difficult to get back to sleep again	3	0	23.08	0
I’m tiring much faster than before	11	6	84.62	46.15
I get tired of everything I do	0	1	0	7.69
Well, I’m getting more tired than before	2	6	15.38	46.15

## Data Availability

Raw data is provided in the Appendix A.

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
