# Peer review of "The Effect of Vinyasa Yoga Practice on the Well-Being of Breast-Cancer Patients during COVID-19 Pandemic"

_ijerph, 2023, doi:10.3390/ijerph20043770_

Round 1
Reviewer 1 Report
This is a worthwhile topic of study, and I commend the authors for undertaking it. While the study’s results are encouraging in supporting the use of approaches like Vinyasa yoga, there are, I think, a number of issues that raise concerns about the conclusions.
Does this journal usually include the theoretical framing and literature review in the discussion section? In other journals I am familiar with, these usually appear after the introduction.
I would suggest including more information about the survey in the “Questionary form” section, rather than leaving this information until the results section.
You might indicate how long after the program was completed the follow up (post-intervention) questionnaire was administered.
The treatment protocol included the vinyasa yoga asanas, ending with Shavasana; during the period of Shavasana there was bibliotherapy, music therapy, mindfulness rotation training and visualization. Might these alone, without the asanas, been responsible for the positive impacts? Was it that the patients simply had some social interaction with others during a stressful time of isolation? We cannot say for sure, and it not unreasonable to think that either (or both) the interaction with others or the combination of bibliotherapy, music, etc. themselves had a significant impact. In the discussion section, the authors note the significance of reducing social isolation. On lines 320-322, the authors write “Researchers point to the key role of social support in initiating physical activity and regularity [49,50] Their findings provide an important argument for encouraging exercise among people at risk of loneliness due to isolation.” Yes, encouraging exercise might be beneficial, but it may be that the social interaction is the most important aspect of the treatment and not the physical activity, per se. I think this needs to be mentioned in the discussion.
While the discussion of the benefits of yoga in lines 331-375 is worthy, the results of this study cannot conclusively demonstrate that yoga asanas or Vinyasa yoga in particular were beneficial: again, the social aspects of the program may have been responsible for the benefits.
I would suggest expanding the limitations section. You might mention again that not all participants completed the program and that the small sample size also reduces the significance of the statistical findings. For example, due to the small sample size, we wouldn’t be able to determine if any of the demographic or SES factors would be significant. I would add that the lack of any control groups further reduces the ability to conclude that the vinyasa yoga component or even all the components of the program were responsible for the improvements. I would suggest two controls groups for future research: one where participants just spent time together (online or not) and one where they had the treatment protocols without the vinyasa yoga asanas.
A follow-up study would be warranted to determine if the effects are long-lasting.
‘cray’ in line 271—should that be ‘cry’?
In line 272: ‘Participants decried a lesser tendency toward nervousness ….’ I would suggest replacing ‘decried’ with ‘noted’; ‘decried’ means denounced, condemned, criticized.
Need a period after ‘[49,50]’ in line 322.
Author Response
Dear Reviewer,
Thank you very much for your valuable comments.
I send the answers in the attachment.
Best regards

Author Response

(The authors gave the same response as above.)

Round 2
Reviewer 2 Report
Dear Authors,
thank you for the comments and revisions, according to my expertise now the manuscript is ready for publication.